# Reliable and Low-Power Communications System Based on IR-UWB for Offshore Wind Turbines

Aitor Guisasola [1], Ainhoa Cortés [1,2,*], Javier Cejudo [1], Astrid da Silva [1], Markos Losada [1] and Paul Bustamante [1,2]

1   CEIT-Basque Research and Technology Alliance (BRTA), Manuel Lardizabal 15, 20018 San Sebastián, Spain; aitorguisai@ceit.es (A.G.); jcejudo@ceit.es (J.C.); adasilvad@ceit.es (A.d.S.); mlosada@ceit.es (M.L.); pbustamante@ceit.es (P.B.)
2   Department of Electronics and Communications, Universidad de Navarra, Tecnun, Manuel Lardizabal 13, 20018 San Sebastián, Spain
*   Correspondence: acortes@ceit.es; Tel.: +34-943212800

**Abstract:** In this paper, we propose the design of a low-power wireless sensor network architecture that enables robust communications inside offshore wind turbines. This research work is in the scope of the WATEREYE EU Project, where we have designed a corrosion monitoring solution to work unattended. The architecture is composed of several fixed sensor nodes, one mobile sensor node, several anchors and the WATEREYE Computer (WEC). Our approach is based on Impulse Radio Ultra wideband (IR-UWB) technology offering reliable and low-power communications in these harsh environments. On top of that, we propose a double star network using two UWB channels for the following purposes: one network for communications to send the sensor data and another one for ranging estimations to calculate the indoor positioning of the mobile sensor node. The power strategies applied to our system, at Hardware (HW) and Firmware (FW) levels, are described in detail. Furthermore, we present power consumption measurements obtaining the power profiles and the autonomy of the most important components of the proposed architecture supplied by battery. On the other hand, we describe the methodology to analyze the range, reliability and continuity of the two UWB links providing the packet loss and gaps as a function of distance. The proposed communications system has been validated in three different scenarios considering two of them very hostile environments. Furthermore, one of the scenarios is a real offshore wind turbine.

**Keywords:** harsh environment; low power consumption; wireless sensor network; IR-UWB; offshore wind turbines

## 1. Introduction

Offshore wind energy is one of the fastest-growing energy sources globally due to its ability to provide power in a stable and predictable way at a very competitive price. Given the European Commission's proposal to raise the European Union (EU)'s renewable energy target for 2030 from 32% to 40%, reducing the actual levelized cost of energy (LCoE) in offshore wind farms (OWF) is one of the key challenges for the decade ahead [1].

In order to achieve the EU's 2030 Climate Target Plan, it is necessary to reduce the operation and maintenance (O&M) costs in OWF. Nowadays, those O&M costs can represent up to 30% of the mentioned LCoE, not only due to the challenging access to the wind turbines (WT) and the adverse environmental conditions at the sea, but also because of the poor O&M strategies that have been carried out traditionally. A typical example of inefficient planning would be a strategy that relies on the historical failure times of the WT components [2].

In contrast, the development of smart O&M strategies based on remote monitoring solutions can greatly reduce downtimes for unplanned maintenance and machinery failures. In fact, those monitoring systems can provide a continuous view of the state of the platform, giving the operators the chance to organize the costly visits to the WTs efficiently [2,3]. Considering that corrosion is the main root cause for offshore structural failures, developing

corrosion-related structural health monitoring (SHM) solutions is essential to recognize early signs of structural risks. In that regard, this paper presents the design of a wireless sensor network (WSN) architecture that enables reliable communications inside offshore wind turbines (OWT) so that the level of corrosion at critical points of the tower can be monitored remotely.

A wireless architecture is the only feasible option for our monitoring system because, unlike other wired alternatives, it enables cost-effective installations in remote and hostile areas for either temporary, mobile or permanent applications [3–5]. However, wireless sensors are not, exactly, cable-based sensor replacements. Without wires, the nodes usually depend on internally stored power for operation. Considering that nodes inside WTs are usually deployed in very tight and enclosed spaces, the use of a small battery for space and weight savings is the most common approach [2,5]. Two of the main factors that impact on the time before a battery replacement is required are presented below:

- Sleep/wake-up mechanism: sensor nodes apply low-power technology to minimize power consumption while ensuring the expected operation by the monitoring system in place. Applications achieve low-duty cycles thanks to the ability of the nodes to wake-up, generate data, transmit them and go back to sleep in a very short time [3,6]. Nowadays, advanced ultra-low-power microcontrollers require current supplies below 1 µA in deep-sleep mode and they can transition to run mode in less than 100 µs [7].
- Network traffic: considering that each extra feature among a WSN (computational ability, self-organizability, information-fusion, etc.) requires power consumption, the processing capacity of the network usually clashes with the low-power requirements of the application [6]. However, pre-processing data onboard, right at the edge of the network, is more efficient than transmitting all the raw data to the cloud. Reducing the size of information reduces the energy required by the radio transceivers to transmit/receive those data. At the same time, throughput, latency and reliability are enhanced by the diminished network traffic, avoiding packet collisions and the consequent re-transmission costs [8,9].

  The type of network traffic also impacts on the battery lifetime. In that regard, a WSN may operate in a beaconing or beaconless mode. The use of beacons is appropriate for real time applications while the beaconless system is useful for applications where the nodes activate just to report the occurrence of an event [10]. The beaconing mode can manage periodic data, where the node wakes up, checks for the beacon, exchanges information and returns to deep-sleep. Furthermore, when a communication without contention or latency needs to be ensured, repetitive low latency data can be handled by allocating a guaranteed time slot to each node. Finally, intermittent data are handled in a beaconless way or in a disconnected manner, where the node joins the network only when it needs to communicate [3].

One of the most important decisions when designing a WSN is the selection of the radio technology (or technologies) used in the wireless links. Several approaches based on the IEEE 802.15.4 standard have emerged as suitable options for the development of low-power low-cost networks [3,11,12].

ZigBee PRO is an enhancement of the original ZigBee protocol stack controlled by the Connectivity Standards Alliance. ZigBee improves the functionality of the IEEE 802.15.4 standard by providing flexible network topologies, intelligent message routing and enhanced reliability. WirelessHART is one of the most widely spread standards within industrial automation. Based on the HART protocol, it implements a synchronized, self-organizing and self-healing mesh routing topology [11,13]. ISA100.11a standard shares many properties with WirelessHART. Both use similar mechanisms to form the network and to transport data to and from the base station. However, WirelessHART is focused on addressing end-user concerns as security, whereas ISA100.11a is designed to provide extended build options for industrial WSNs [14].

Impulse Radio Ultra-wideband (IR-UWB) is also considered a feasible technology for WSNs due to its excellent time-domain resolution and its low-power low-cost features [15]. It should be noted that it is based on the IEEE 802.15.4a standard, the first international standard that specified a wireless physical layer to enable precision ranging at low power density. This standard was designed to deliver very high accuracy location while providing simultaneous two-way communication—up to 27 Mbps—to collect sensor data and control actuators.

UWB uses very short pulses in the time domain, which implies that in the frecuency domain, as its name suggests, the signal is much wider than in other communication systems [16]. Due to the combination of wide bandwidth and low-power, UWB signals are robust under interference and do not interfere in other systems. On the ond hand, the wide bandwidth protects UWB against interferences from narrowband systems as well as from multi-path effects. On the other hand, the low-power spectral density enables coexistance between UWB and narrowband systems [15].

In that regard, UWB systems are characterized by operating in the presence of frequency selective receivers with low interference, which enhances the robustness of the link [17]. Measurements on offshore oil platforms, where conditions are comparable to those on OWTs, have shown that UWB is a suitable option for environments where large metal components or electromagnetic devices interfere [11,17]. Besides, UWBs data rate is considered acceptable for applications where the nodes just transmit small messages for configuration purposes or to exchange the acquired data [11].

To cope with the diversity among SHM systems, several network topologies are used in WSNs. The star and mesh networks have been considered for our WSN design:

- Star network: a base station is in charge of sending/receiving the messages to/from every remote node. At the same time, those nodes are allowed to send/receive message only to/from that single base station. The main advantages of the star network are its simplicity and the energy savings resulting from the single-hop routing system. On the other hand, the network has absolute dependency on the base station and every node must be within its radio range [6,12].
- Mesh network: any node is allowed to send/receive messages to/from any other node in the network. Consequently, the network can recover from a broken node by reconfiguring the blocked paths and hoping from one node to another. However, as the number of communication hops increases, the overall power consumption of the network also increases. Apart from the mentioned redundancy, multi-hoping also provides scalability. In that sense, the network range can be easily extended by adding extra nodes [6,12].

As the power consumption and robustness are the key challenges of our approach, we propose a double star network topology based on IR-UWB applying low-power strategies at HW and FW levels. Thus, our system not only employs synchronized sleep/wake-up mechanisms to minimize power consumption while ensuring a reliable data communication, also presents a HW design capable of disabling the components of the system in a controlled manner to increase the battery lifetime even more. On the other hand, we have designed a proprietary protocol taking into account these two challenges and looking for a trade-off between the processing in the edge and the size of the payload to be sent wirelessly.

The paper is structured as follows. In Section 2, a review of different WSN architectures in harsh environments is introduced. Section 3 provides the description of the case study of this research work. Section 4 presents the implementation of the proposed WSN architecture. In Sections 5 and 6, the test setup, the measurement methodology, and the experimental results are presented. Finally, the discussions and conclusions obtained from the experimental results are summarized in Section 7.

## 2. Related Work

The development of smart SHM systems for OWTs is already playing a huge role in the establishment of offshore wind energy as the most profitable renewable energy source. Bearing in mind that each monitoring application has its own needs, different approaches have been found in the literature in terms of the implemented architecture, communication technology and/or power-saving strategy. However, it should be mentioned that every reviewed monitoring solution is based on wireless communication links [2–4,8,9,11,12,18,19].

A challenge when implementing a WSN is to select the technology that best suits the different requirements of the communication system and its application. Table 1 shows a comparison between different technologies considered for low-power WSN. The IEEE802.15.4 standard has been designed to achieve low data rate, low power consumption and low cost. WIFI halow is a good option to have a medium range and high data rates. Bluetooth Low Energy (BLE) is considered a good alternative when short ranges and medium data rate demands are required. LoRaWAN is an excellent alternative to cover long distances. If we focus on the ranging estimation capabilities, UWB achieves the highest accuracy. It must be said that the coverage distance, the ranging accuracy and also the power consumption are very affected by the system configuration, the system architecture and the environmental conditions.

**Table 1.** Comparison between different technologies for WSN.

|  | UWB [11,15] | Zigbee [10,20] | LoRaWAN [9] | Bluetooh BLE [20] | WIFI HaLow [21] |
|---|---|---|---|---|---|
| Standard | 802.15.4a | 802.15.4 | LoRaWAN specification | Bluetooth 4.0 | IEEE 802.11ah |
| Frequency | 3.1–10.6 GHz | 2.4 GHz | 433-868-915 MHz | 2.4 GHz | 2.4–5 GHz |
| Coverage Distance (m) | 200 | 250 | >1000 | 200 | 1000 |
| Maximum Data Rate | 6.8 Mbps | 250 kbps | 50 kbps | 3 Mbps | 54 Mbps |
| Network Topology | Star, Mesh | Star, P2P, Mesh, Tree | Star | P2P, Star | Star |
| Ranging Accuracy | 10–15 cm | meters | N/A | 1–3 m | few meters |

As described in Section 1, WSNs use different topologies to address the unique requirements of each SHM application. For example, Ref. [12] opts for a mixed solution after assessing the pros and cons of the mesh and star network topologies. Specifically, they decide to implement a mesh network for communications between WTs and a star network for communications involving an individual WT.

On another note, Ref. [4] presents a cluster-based distributed topology. In particular, the research work presents an architecture where four cluster nodes are deployed for fusing and transmitting the data collected by several sensor nodes located on strategic parts of the WT. On the one hand, the sensor nodes belonging to the same cluster can communicate with each other in their way of reaching the target cluster. Similarly, the cluster nodes can back each other up when transmitting the further fused data to the control center. This way, the network routes are optimized by a double multi-hoping communication protocol. In that regard, Ref. [18] assesses the potential effects associated with the operation, environment and structure of WTs in multi-hop communication systems and evaluates a gradient-based routing protocol that counteracts those effects.

On the opposite side, Ref. [19] presents a SHM application with real-time requirements in order to guarantee synchronized data acquisition. The paper concludes that the best solution to achieve real-time performance is to adopt a simple star network that exploits time

division for the network access policy. Besides, the research work explains that standard protocols either lack of deterministic features (WiFi, ZigBee) or scalability (Bluetooth) to be considered for industrial monitoring solutions. Therefore, they opt for a similar approach to the industrial standards described in Section 1 (WirelessHART, ISA100.11a), adding the required elements for obtaining an accurate sampling synchronization.

Refs. [3,11,12] assesses the special conditions and requirements for WSNs in harsh offshore environments. The three papers determine that IEEE 802.15.4-based standards are feasible options for those kind of maritime projects. However, Ref. [11] explains their need to find an alternative standard given their goal of covering both data communication and localization in offshore platforms. The research work brings up UWB technology as the best solution to achieve that combination as it enables simultaneous communication and precise distance measurements in moderate ranges. However, they do not present power consumption measurements of their approach. On top of that, they do not validate their proposal in a relevant maritime environment.

Ref. [9] proposes a different approach based on LoRaWAN for outdoor communications, not inside a WT, in which the WSN has the ability to select the most appropriate radio-technology for each type of data communication. The presented multi-technology network takes into account aspects as the amount of information to be sent, the required data rate or the range to be covered in order to pick the best alternative. In that regard, a WSN design with backup radio-technologies could be specially beneficial for harsh environments as OWTs where conditions are highly variable. It must be taken into account that LoRa is adequate for long-range communications with a low throughput.

Finally, previous related work also addresses the problems of designing a WSN in the context of offshore wind farming. Ref. [6] discusses the key issues of WSNs applied in SHM systems while [2] is focused on the development of an efficient and reduced power miniaturized system for corrosion monitoring. It should be noted that both papers identify the strategy for powering the WSN system as a key challenge. Unfortunately, except [9] which proposes a very different approach, none of the proposals in the literature perform real power consumption measurements.

Ref. [2] reminds that both hardware design (e.g., component selection) and software design (e.g., data acquisition strategy) must consider the power and timing requirements for conducting sensor measurements. The research work also presents a set of test results that helped them optimizing the number of data to be collected per measurement. As explained in Section 1, reducing the amount of information transmitted by the nodes reduces the power consumption derived from those communications as well. In that sense, Ref. [8] proposes a fault detection system using adaptive thresholds which is efficient in reducing the size of the data packets transmitted by the sensor nodes. Therefore, the method is also effective in saving energy and enhancing the node lifetime. Ref. [22] proposes a new network monitoring protocol which combining features from other commonly used protocols (SNMP, CoAP) is efficient in reducing the amount of bytes transmitted in the application layer. However, we have opted for an ad hoc approach when designing our proprietary protocol, since it fits best the features of our ultra-low power sensor network.

In summary, we have found different wireless communications systems relying on different technologies and approaches, some of them for industrial environments. The enhancement of the network lifetime is identified as a key challenge, however, there is a lack presenting specific power consumption measurements as well as detailed node lifetime estimations. As explained before, the proposed WSN employs different UWB channels when addressing communication and localization at the same time, which is also an innovative approach to minimize interferences within the network. Finally, we have identified a gap in the validation of WSN solutions in relevant harsh environments, or even in maritime environments as is the focus of this paper. This work presents specific results of coverage measurements in moderate ranges at different harsh scenarios, providing insight for UWB communications in those special environments.

### 3. Case Study: Smart Monitoring System for Offshore Wind Turbines

This research work is focused on the monitoring of Offhsore Wind Turbines as the case study to prove the efficiency of the proposed wireless communications system. The target Wind Turbines are made of steel which is not the best scenario for wireless communications. This means that the wireless link will have to work in a multi-path environment which can considerably degrade its reliability.

This research work has been supported by the WATEREYE Project as is explained in the Funding section. WATEREYE is focused on the design of O&M tools integrating accurate structural health monitoring in offshore energy with the main objective of reducing O&M costs. Hence, WATEREYE aims to deploy a smart monitoring system inside the Offshore Wind Turbines to predict the Remaining Useful Life (RUL) of offshore steel structures. Specifically, this monitoring solution will operate unattended 24/7 during years to monitor the degradation of the tower structure due to corrosion, which has a serious impact over these offshore structures.

The monitoring system is composed of two kind of sensor nodes: a mobile sensor integrated into a drone which will cover the atmospheric zone of the tower and several fixed sensors monitoring specific critical points of the splash zone. All these sensor nodes will have to send the measured data in real-time wirelessly to the base-station located also inside the offshore tower, which is called WATEREYE Computer (WEC). Hence, the design of a robust and reliable wireless network is totally needed.

The aim of the wireless communication system is to achieve these robust communications inside the WT tower between the sensor nodes and the WEC. To do that, proprietary protocols have been developed, instead of using existing standard network stacks, to work in the harsh environments of the identified WATEREYE scenarios without losing data frames or receiving corrupted information and at the same time reducing even more the power consumption. This system will:

- Be based on radio-frequency communications, specifically on UWB technology, which is robust in a multipath environment and permits to achieve a radio link with a very low energy consumption. This is key to achieve the required autonomy of the sensor nodes between 3 and 5 years. Additionally, the UWB technology will allow us to improve the positioning of the mobile (drone) platform, which will fly inside the tower thanks to some important features of the UWB signals as their high band-width they can provide.
- Be robust by applying proprietary communication protocols to ensure reliable communications and low-power consumption. The low-power consumption is a key factor. On the one hand, the system must be totally autonomous and non-dependent on the environmental conditions. On the other hand, the deployment of the system must be easy and cost-effective.
- Have an effective range of 80 m and a maximum data rate of 6.8 Mbps. The effective range of 80 m is considered enough to cover the wireless communications inside the wind turbine tower to connect wirelessly the mobile platform, the fixed sensor nodes and the WATEREYE Computer. On the other hand, a maximum data rate of 6.8 Mbps should be enough to send the quantity of data required in WATEREYE. Considering as well that the corrosion phenomenon is a slow process, we can use a lower data rate in the order of hundreds of kbps.
- Comply with the required timing of 150 ms among ranging estimations and a ranging accuracy of 15 cm to provide the capability of following a trajectory
- Be flexible to work for other applications and other monitoring systems.
- Have low-weight and small size to facilitate the deployment and to be integrated in a mobile platform like a drone.

### 4. WATEREYE System Architecture

The proposed system architecture is shown in Figure 1.

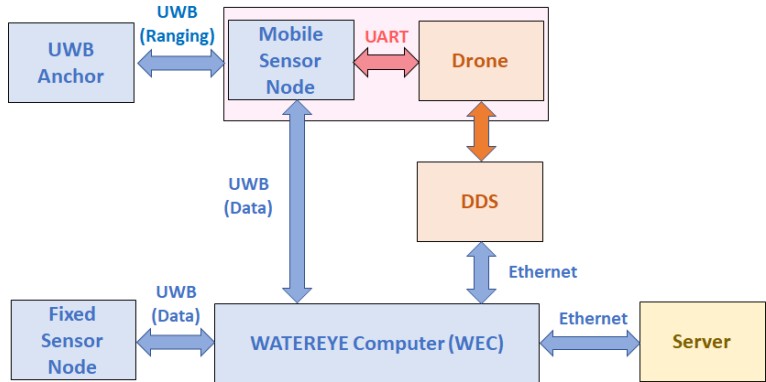

**Figure 1.** WATEREYE system architecture.

The WATEREYE system is mainly formed by five subsystems: the Fixed Sensor Node, the Mobile Sensor Node, the Drone, the UWB Anchor and the WATEREYE Computer. The final solution will have two more modules: the Drone Docking Station (DDS), which will enable the interaction between the WEC and the Drone; and the WATEREYE Server, which will integrate the WATEREYE DataBase and the tools to predict the RUL of the structure. Specifically, this paper is focused on the design of the Sensor Nodes (Fixed and Mobile), the UWB Anchor and the WEC. The architecture and operation principle of these subsystems is explained in the following sections.

As shown in Figure 2, the WATEREYE system architecture implements a double star network. On the one hand, the UWB Anchors only send/receive messages to/from the Mobile Node through UWB Ranging links. On the other hand, the Fixed Nodes only send/receive messages to/from the WEC through UWB Data links. The Mobile Node also communicates with the WEC through an UWB Data link. A star network is suitable for our architecture since every sub-system will be deployed inside a wind turbine. Therefore, the WEC will be within the radio range of all Sensor Nodes and the Mobile Node will be within the range of all Anchors. Besides, the dependency of the network on key sub-systems as the WEC and the Mobile Node is a reasonable price to pay for having a simple and low-power consumption network.

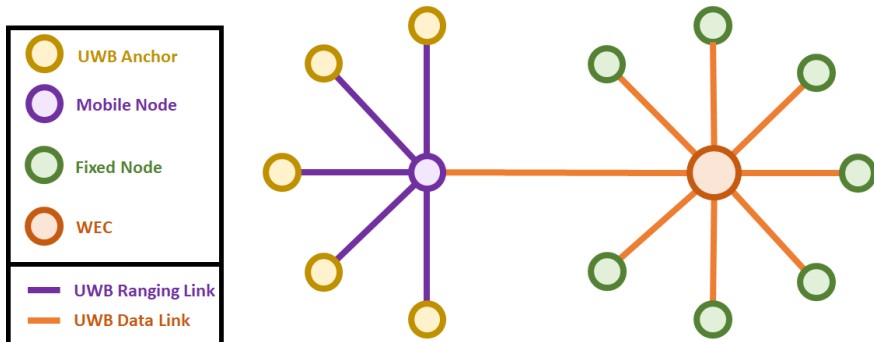

**Figure 2.** WATEREYE system network topology.

*4.1. The Sensor Nodes*

- **Hardware**
  Figure 3 shows the architecture of the Sensor Node.

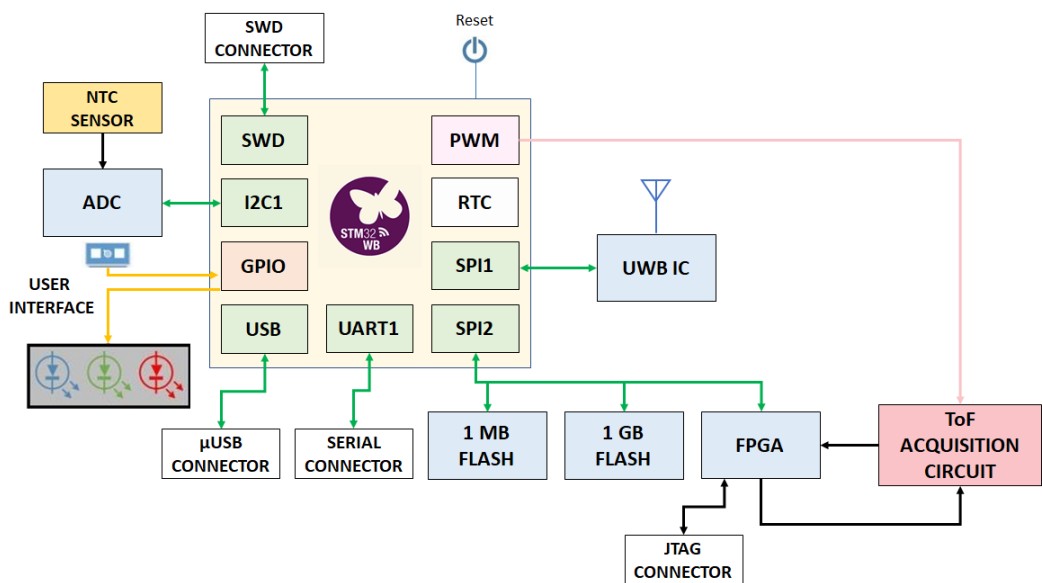

**Figure 3.** Sensor Node architecture.

A custom board has been designed in order to support this architecture to fulfill the low-cost, low-weight and small size requirements. Furthermore, we will be able to apply low-power strategies at HW level enabling the integrated circuits only when necessary. The STM32WB55 microcontroller is the core of processing and control. This microcontroller incorporates an ARM based Cortex-M4 32-bit core operating at a 64 MHz frequency, 1 Mbyte of Flash memory, 256 Kbytes of SRAM and a low-power real-time clock (RTC). It can be programmed through an SWD interface and a micro USB connector provides access to a virtual com port. The STM32WB55 also features standard and advanced communication interfaces as two UARTs, two I2Cs and two SPIs [7].

Apart from the microcontroller's internal memory, the Sensor Nodes contain a 1 MB Flash memory for storing configuration parameters and a 1 GB Flash memory that allows us to store corrosion measurements when desired. Both of them are accessed through a SPI bus. The Sensor Node also houses Intel's MAX 10 FPGA, which is accessed through the same SPI bus. When the microcontroller requests a Time-of-Flight (ToF) measurement, the FPGA takes control of the devices that form the data acquisition circuit. Once the measurement is performed, the FPGA processes the acquired data and provides the calculated ToF estimation to the microcontroller. The Sensor Node also reads from a 16-bit analog-to-digital converter through I2C to link temperature values to the corrosion measurements. The variation ramp of the ToF with temperature will be considered in further processing.

As mentioned before, IR-UWB technology is provided to the designed board by means of the DW1000 [23] Integrated Circuit (IC) from Decawave, an IEEE802.15.4-2011 UWB compliant low-power and low-cost wireless transceiver. A SMA connector has been deployed to use an omnidirectional UWB antenna. The STM32WB55 interacts with the UWB IC using a separate SPI bus.

In this architecture, both the Fixed Sensor Nodes and the Mobile Sensor Node use UWB technology to send the corrosion measurements to the WEC. The Mobile Sensor Node also employs UWB to make precise ranging estimations with the UWB Anchors. The IEEE 802.15.4 UWB PHY has 16 defined channel/bands of which channels number 1, 2, 3, 4, 5 and 7 are supported by the DW1000 IC. Besides, the IC supports data rates of 110 kbps, 850 kbps and 6800 kbps. Table 2 shows the main parameters of the system configuration. It is important to highlight that our architecture implements two different channels at two different data rates according to the purpose of the communication: send data or make ranging estimations. This way, the risk for

interferences within the network is reduced. A maximum 6800 kbps data rate is used in the ranging links in order to achieve a low ranging time by speeding up the distance estimation process. On the other hand, a 110 kbps data rate is enough for the data links since the transmitted payload is small and the communications happen occasionally.

**Table 2.** UWB links in the WATEREYE system architecture.

| Parameters | UWB Link | |
|---|---|---|
| | Data | Ranging |
| Channel number | 3 | 4 |
| Centre frequency (MHz) | 4492.8 | 3993.6 |
| Data rate (kbps) | 110 | 6800 |
| Bandwidth (MHz) | 500 | 800 |
| Preamble Length (Sync.) | 1024 | 128 |
| Power Spectral Density (dBm) | −41.3 | −41.3 |

In the case of the Fixed Sensor Nodes, a primary battery of 3.6 V and 5800 mAh provides the power supply. On the opposite side, the Mobile Sensor Node is powered by the Drone. In both cases, the Sensor Node generates several voltages between 15 V and 1.8 V to supply the devices that form the subsystem. However, the only supply that remains always active is the 3 V supply that powers the microcontroller. This way, STM32WB55's internal RTC is able to wake-up the Node by triggering a pre-configured alarm. The rest of the voltage supplies stay active only when they are required for performing the tasks related to the corrosion measurements or communication events. Figure 4 shows a photo of the designed board. Mechanical dimensions are 110 × 60 × 15 mm.

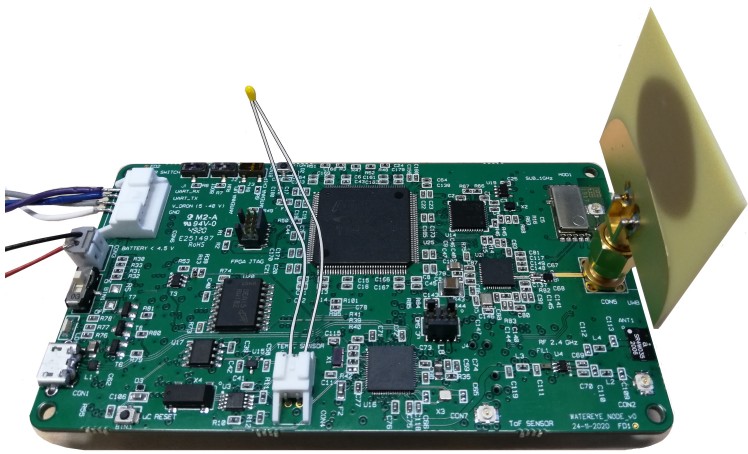

**Figure 4.** Sensor Node board.

- **Firmware**
  Considering that a battery autonomy of 3–5 years is required, the Fixed Sensor Node must reduce its power consumption by decreasing its duty cycle. In our application, that duty cycle is determined by the periodicity of the two main events performed by the Fixed Sensor Node: Beacon event and Measurement event. The behaviour of these events is described by means of Figure 5 where the flow chart of the implemented FW in the microcontroller is shown.

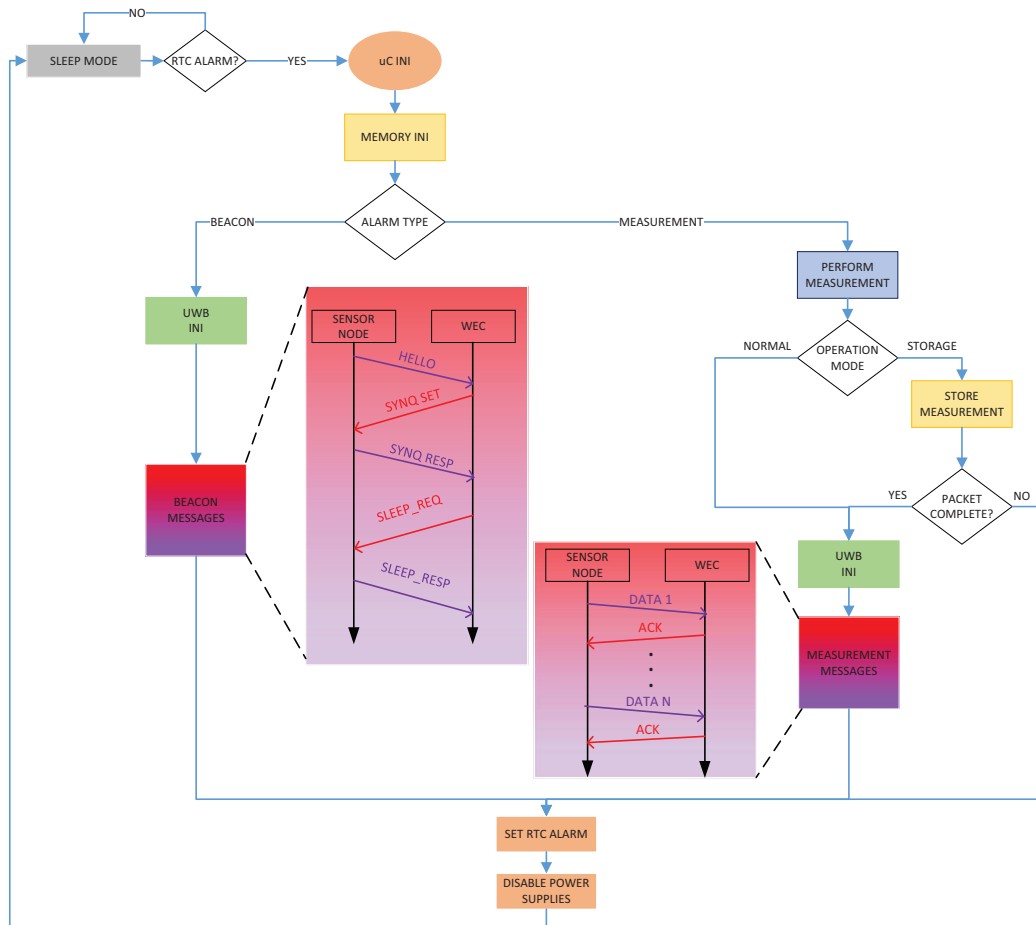

**Figure 5.** Sensor Node flow chart.

In the case of the Beacon event, the Sensor Node performs the following tasks after being waked-up by the RTC: initializes the microcontroller, loads its configuration parameters from the external 1 MB Flash memory, configures the UWB interface, performs a message interchange with the WEC for synchronization/configuration purposes, sets-up the next alarm, disables the power supplies and goes back to deep sleep. Each Fixed Sensor Node will have a specific wake-up time-slot to minimize collisions between beacons from different Nodes.

As far as the process of performing and transmitting the corrosion measurements, the Node has two operational modes. In Normal mode, the Node makes a corrosion measurement and immediately sends the acquired data to the WEC without storing it internally. In Storage Mode, the Node stores the measurements in the external 1 GB Flash memory and only sends the complete set of stored measurements once every several cycles. The cycle in which the Node makes a measurement and stores it in memory is called Measurement event. The cycle in which the Node measures and transmits the whole packet is called Measure & Transmit event. Again, each Fixed Sensor Node will have an specific wake-up time-slot to perform and/or send the measurements. However, re-transmission is an unavoidable event in a WSN, especially when the system is under harsh environmental conditions. Therefore, we have implemented a re-transmission protocol to improve the robustness of the data transfer from the Nodes to the WEC.

Taking into account that UWB frames can contain up to 127 bytes, we can send a maximum 4 corrosion measurements in each data frame. When more than 4 measurements need to be transmitted, the whole data set will be fragmented in several sub-frames. Each frame will have a header including the index of the frame and the maximum index of the data set. This way the WEC will be able to build the whole message.

The WEC responds to each frame with an acknowledgement (ACK) message to inform the Node that it has received the information properly. If an ACK message does not arrive within 20 ms, the Node re-transmits the specific sub-frame up to 10 times. It should be noted that the header of the ACK message also includes the ID of the target Node and the index of the message which is acknowledging to avoid miscommunications between the Nodes and the WEC.

### 4.2. The UWB Anchor

- **Hardware**
  Figure 6 shows the architecture of the UWB Anchor.

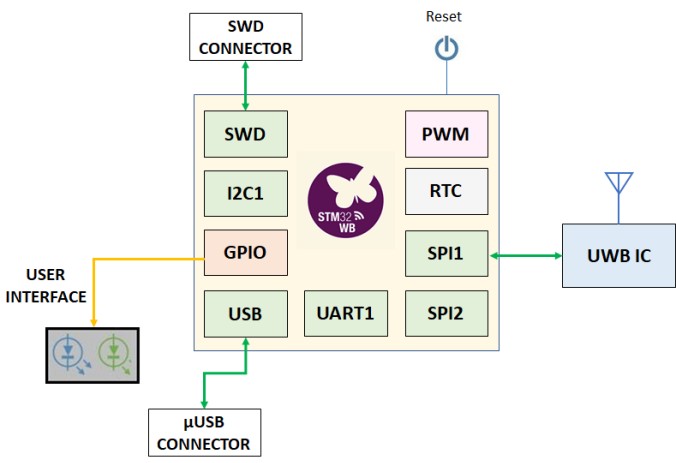

**Figure 6.** UWB Anchor architecture.

A custom board has been designed in order to support this architecture to fulfill the low-cost and small size requirements and facilitate the deployment. Furthermore, we will be able to apply low-power strategies at HW level enabling the integrated circuits only when necessary. The UWB Anchor is identical to the Sensor Node as far as the employed microcontroller (ST's STM32WB55), UWB IC (DW1000) and power source (3.6 V and 5800 mAh battery). Similarly, it can be programmed through an SWD interface and a micro USB connector provides access to a virtual com port through an UART interface. An RGB led has been included to act as a visual interface for the user. The main regulator is a Low-DropOut (LDO) regulator which feeds the microcontroller at 3.3 V. This regulator is always active and hence the microcontroller too. Another LDO (LDO1) regulator feeds the DW1000 IC at 3.3 V which can be enabled by the microcontroller.

Figure 7 shows a photo of the designed board. Mechanical dimensions are 85 × 62 × 15 mm.

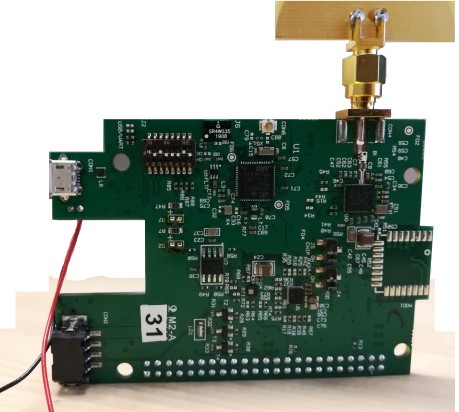

**Figure 7.** UWB Anchor board.

- **Firmware**

  In order to achieve a battery autonomy of 3–5 years, the Anchor must decrease its duty cycle. In our application, that duty cycle is determined by the periodicity of the two main events performed by the Anchor: Beacon event and Ranging event. The behaviour of these events is described by means of Figure 8 where the flow chart of the implemented FW in the microcontroller is shown.

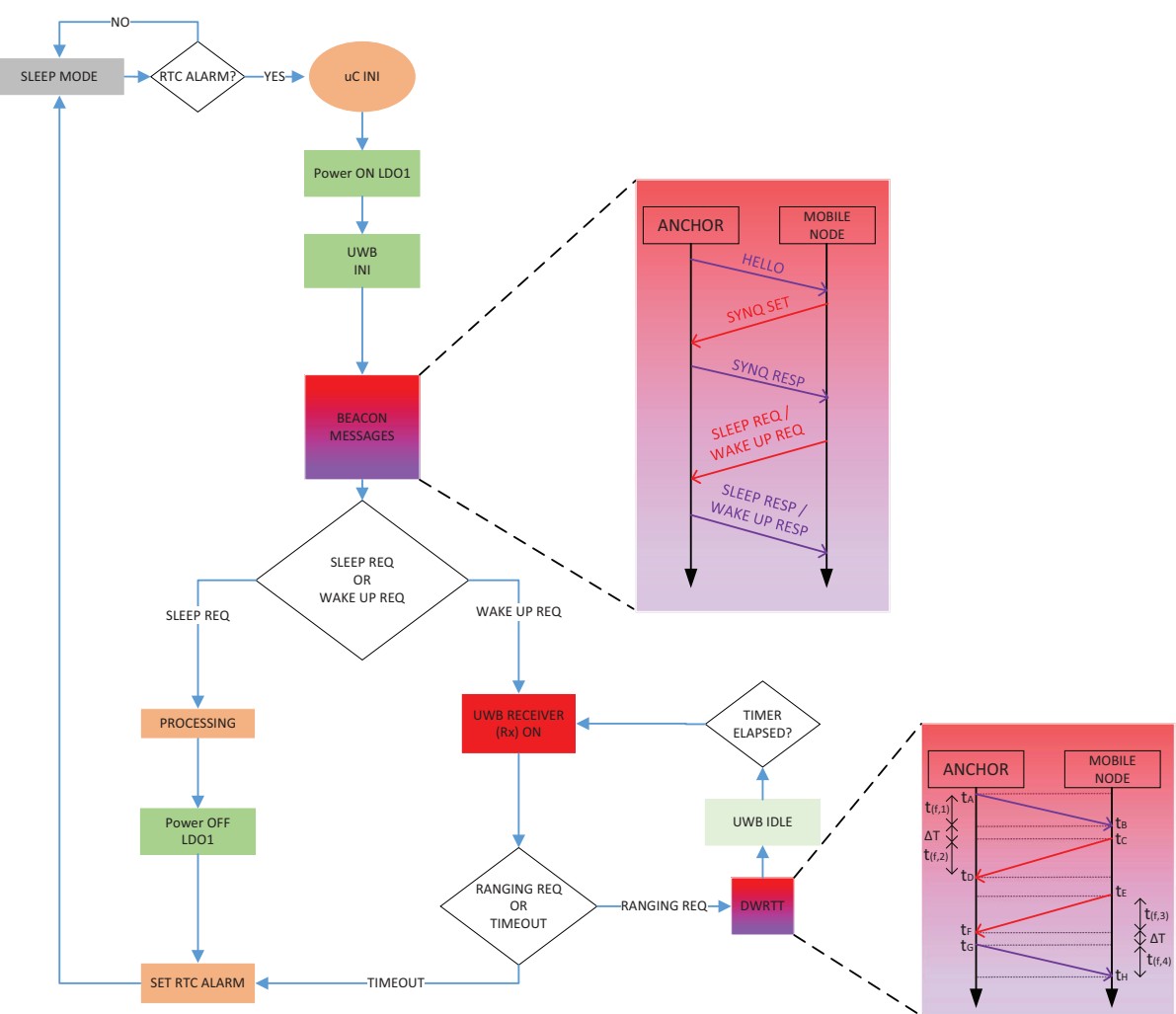

**Figure 8.** UWB Anchor flow chart.

Once the microcontroller is waked-up by STM32WB55's internal RTC, it enables the LDO1 that feeds the DW1000 IC and then configures the UWB communications. It should be noted that each Anchor will be pre-configured with an specific wake-up time-slot so that collisions between Anchors are minimized. Those time-slots will be based on the Anchor ID. Next, the Anchor and the Mobile Node interchange beacon messages. At this point, the communication can lead to two situations. The regular Beacon event corresponds to the situation in which the Mobile Node has no queries for the Anchor, thus the Node sends a sleep request to the Anchor. Therefore, the Anchor disables the LDO1, reconfigures the RTC alarm with the specified sleep time and it goes back to the Sleep mode. On the other hand, the Ranging event corresponds to the situation in which the Mobile Node sends a wake-up request. In this case, the Anchor will turn on the UWB receiver to listen to the Mobile Node's request and start the ranging process. If the Anchor does not receive the request within an specified timeout, the Anchor will return to Sleep mode.

Round-trip-time (RTT) is the time it takes for a signal or message to travel from a transmitter to a receiver and back again. RTT measurements do not require absolute clock synchronization. The Anchor calculates the ToF between the Anchor and the Mobile Node by following Equation (1):

$$t_f = \frac{(t_B - t_A) + (t_D - t_C)}{2} = \frac{t_{(f,1)} + t_{(f,2)}}{2} = \frac{t_D - t_A - \Delta T}{2} \tag{1}$$

The Double Way Round-Trip-Time (DWRTT) technique performs the RTT method twice to eliminate the clock drifts among Anchors and the Mobile Node. Figure 8 shows how firstly an RTT estimation from the Anchor to the Mobile Node is performed, and after that, another RTT estimation from the Mobile Node to the Anchor is carried out. Thus, ranging measurements are taken by both the Mobile Node and the Anchor. Then, these two ranging estimations are averaged by the Anchor. In this case, the ToF between the Anchor and the Mobile Node is calculated by following Equation (2):

$$t_f = \frac{(t_{(f,1)} + t_{(f,2)}) + (t_{(f,3)} + t_{(f,4)})}{4}. \tag{2}$$

### 4.3. The WATEREYE Computer

A block diagram of the WATEREYE Computer is presented in Figure 9.

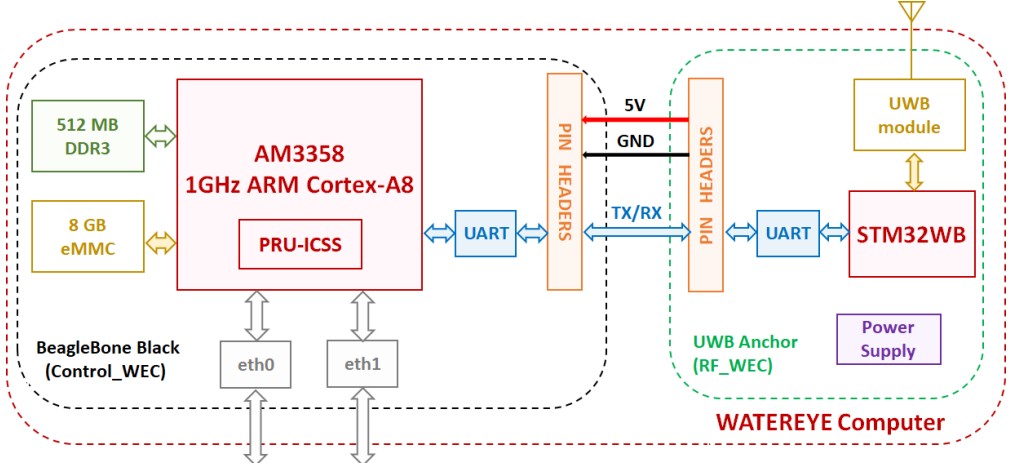

**Figure 9.** WATEREYE Computer architecture.

The WEC is formed by a BeagleBone Black (BBB) development platform, which works under Debian GNU/Linux and whose main component is TI's AM3358, an up to 1 GHz Sitara ARM Cortex-A8 32-bit RISC processor. Among its main features, the BBB includes a 512 MB DDR3 RAM, a 4 GB eMMC on-board flash storage and two Programmable Real-Time Units (PRUs) capable of running at 200 MHz. The board also incorporates 92 GPIO pins which can be configured in 8 possible operation modes by loading customized Device Tree Overlays, providing great flexibility to the user [24].

The corresponding HW to an UWB anchor (RF_WEC) is attached to the BBB (Control_WEC) through an UART interface in order to provide UWB connectivity to the WEC. When the RF_WEC sub-module receives a message through its UWB port, the RF_WEC bypasses the message to the Control_WEC. In the same way, after processing the received message and building the corresponding answer, the Control_WEC transmits the message through UART to the RF_WEC. Immediately, the RF_WEC sends the message wirelessly through UWB to the target Sensor Node.

Additionally, the BBB includes an Ethernet port and a Type-A USB 2.0 host port, which will be used to set up the Ethernet interfaces with the WATEREYE Server and the DDS. On the one hand, the WEC will send the measurement data in JSON format to the WATEREYE Server on the cloud through HTTP requests. On the other hand, the WEC will

communicate through TCP/IP protocol with the DDS to provide the list of points which define the trajectory to be made by the Drone.

The WEC will be connected directly to the power network so that low-power strategies do not apply in this case.

## 5. Power Consumption Measurements

### 5.1. System Setup

Figure 10 shows the setup proposed to measure the power consumption generated by the Sensor Node and the UWB Anchor. The same setup configuration has been implemented for both subsystems. The Otii Arc [25] power analyzer and power supply have been used to set a 3.6 V power supply and also monitor the current consumption. This tool has been used as a profiler to measure currents with a high input range. It can measure magnitudes from 5 amps down to tens of nanoamps with a sample rate up to 4 ksps. A desktop application has been connected to the Otii Arc USB port in order to record and display the measured currents in real-time. Moreover, we have synchronized the debug logs of the Node/Anchor with the current and voltage graphs displayed by Otii Arc. This way, it is possible to correlate the power profile to each application state.

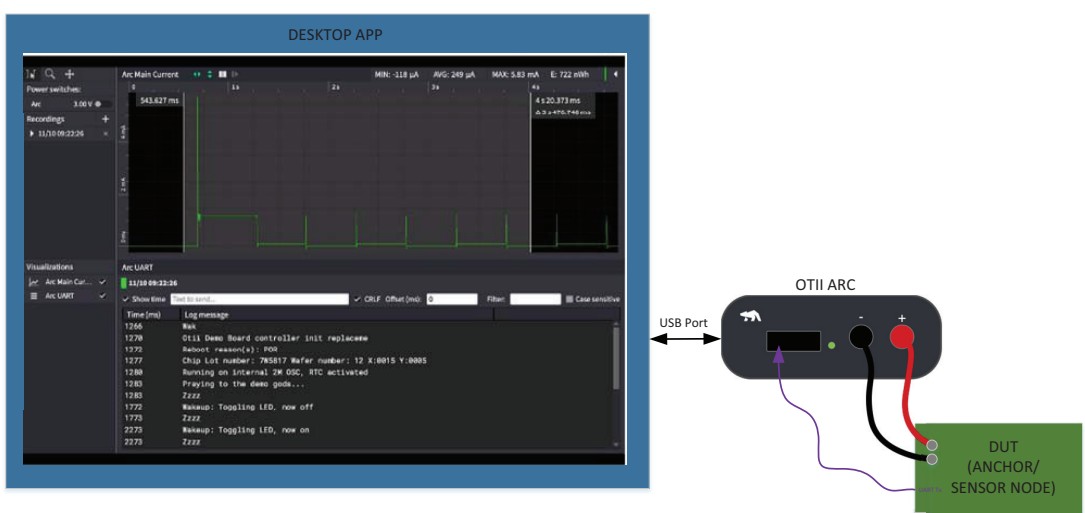

**Figure 10.** Implemented setup for the power consumption measurements.

### 5.2. Methodology

As stated in Section 4, different modes and events exist during the operation of the Sensor Nodes and the Anchors. The average power consumption for each type of event ($\overline{P_{ON}}$) has been measured as well as the average power consumption at Sleep mode ($\overline{P_{OFF}}$). The average power consumption for each event is estimated by the power analyzer which is able to calculate the average current of the captured event during a specific period of time.

The consumption profiles of the events generated by the Sensor Node are shown in Figure 11a while the consumption profiles of the events generated by the Anchors are in Figure 11b. These consumption profiles are related to the flow charts of the sensor node and the anchor shown in Figures 5 and 8, respectively. Note that we have used the same coloured codes as in the flow charts highlighting the power profiles of the different stages of the devices under analysis. These profiles are obtained in a qualitative way. Their aim is to show how the different type of events look like in terms of power and time, and get a quick visual comparison of the different tasks within each event. It can be seen that the consumption of the microcontroller in active mode is present all the time in any of the events. On the one hand, in the case of events in which there are communications, it is necessary to turn on the UWB transceiver and hence the consumption increases. This

consumption increases even more when there are UWB transactions. In addition to that, the consumption is higher when the transceiver is in reception mode. On the other hand, in the case of the events of the sensor node, consumption increases since it is necessary to turn on the circuits related to corrosion measurements such as memory, FPGA, ADC or the pulser. Furthermore, in all the cases, except for the ranging event, once the event is over, the device goes to the sleep mode, reducing its consumption considerably.

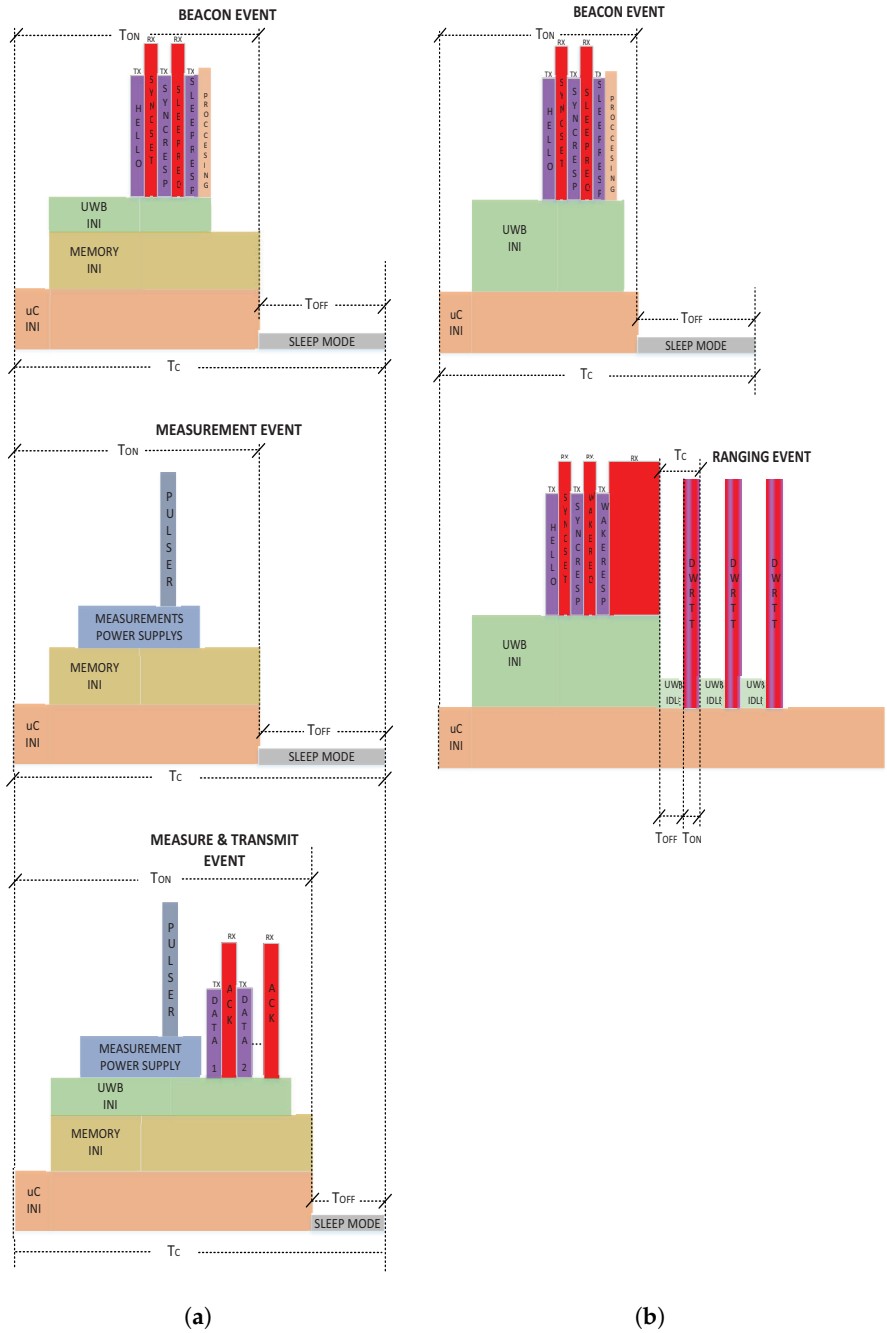

(**a**)        (**b**)

**Figure 11.** Sensor Node and Anchor power profiles. (**a**) Power profile for Sensor Node events. (**b**) Power profile for Anchor events.

The duty cycle is the relationship between the on-time ($T_{ON}$), the time in which the Node/Anchor is in Run mode, and the cycle time ($T_C$), which is the total time of one cycle. The off-time $T_{OFF}$ is the time in which the Node/Anchor is in Sleep mode. The obtained average power consumption ($\overline{P}$) is shown as a function of the cycle time ($T_C$) by following Equation (3), such as

$$\overline{P} = \frac{(T_{ON} \times \overline{P_{ON}}) + (T_{OFF} \times \overline{P_{OFF}})}{T_C} \tag{3}$$

where $T_{ON}$ and $\overline{P_{ON}}$ are the time duration and average power consumption of the corresponding event, respectively. $T_{OFF}$ and $\overline{P_{OFF}}$ are the time duration and average power consumption of the off-time, respectively. For the purpose of this study, we consider that in each operation mode only events related to that operation mode are generated. For example, the mode in which only Beacon events are generated is referred as Beacon mode.

*5.3. Sensor Node Measurements*

Table 3 shows the average power consumption and the time duration for the on-time and off-time for each type of mode. Note that, in the case of the Beacon and Measurement events, the on-time ($T_{ON}$) keeps almost constant for each event. This can be said if we assume that the microcontroller clock frequency is stable with temperature and the UWB communications have the same time duration during the Beacon event. In the case of the Measure & Transmit Event, the on-time ($T_{ON}$) is variable since it depends on the number of consecutive frames to be sent.

**Table 3.** Sensor Node: average power consumptions and time durations for the different modes.

| Mode | $T_{ON}$ (ms) | $\overline{P_{ON}}$ (mW) | $\overline{P_{OFF}}$ (mW) |
|---|---|---|---|
| Beacon | 383 | 388.8 | 0.0054 |
| Measurement | 1000 | 849.6 | 0.0054 |
| Measure & Transmit | variable | variable | 0.0054 |

Figure 12 shows the average power consumption ($\overline{P}$) of the Sensor Node in the Measurement mode and in the Beacon mode as a function of the cycle time ($T_C$). As can be seen, the longer the cycle time, the lower the average power consumption since the sensor node remains more time in sleep mode. Therefore, it can be seen that both curves tend to $\overline{P_{OFF}}$. On the other hand, as expected, for the same cycle time ($T_C$) the power consumption is higher in Measurement mode than in Beacon mode.

From Equation (3), we can observe that ($\overline{P}$) is completely influenced by the off-time $T_{OFF}$. In fact, we can observe that as long as $T_C$ increments, $\overline{P}$ tends to be equal to $\overline{P_{OFF}}$.

It can be seen that in Beacon mode and Measurement mode the battery life is mainly determined by the cycle time. In other words, the longer the cycle time, the longer the time spent in Sleep mode and, therefore, the lower the average energy consumption per cycle.

For the WATEREYE use case, if we consider that the Sensor Node remains in Beacon mode during the whole battery life cycle, then, with a 3.6 V and 5800 mAh battery (20.88 Wh), by setting a cycle time of 4 min, a battery life of 3.8 years is estimated taking into account the measured average power consumption of the sensor node for that cycle time as is highlighted in Figure 12.

Figure 13 shows the average power consumption ($\overline{P}$) by the Sensor Node in the Measure & Transmit mode as a function of cycle time ($T_C$). In this case, the on-time is related to the amount of data transmitted and thus, we can observe different curves for each number of transmitted data frames.

The number of frames to be transmitted depends on the number of measurements to be sent. As explained before, each UWB frame can contain up to 4 measurements. Therefore, if 1 to 4 measurements need to be sent, a single data frame will be employed. If 5 to 8 measurements need to be transmitted, the measurements will be divided into two sub-frames, and so on. As can be seen in Figure 13, for the same duty cycle, the power consumption increases if a higher number of frames is sent. This happens not only because more bytes are sent in the same cycle, but also because of the power consumption related to the reception of the ACK message corresponding to each frame.

On the other hand, as in the case of the Beacon and Measurement modes, for every number of transmitted data frames the longer the cycle time, the lower the average power consumption. Therefore, it can be seen that all the curves tend to $\overline{P_{OFF}}$.

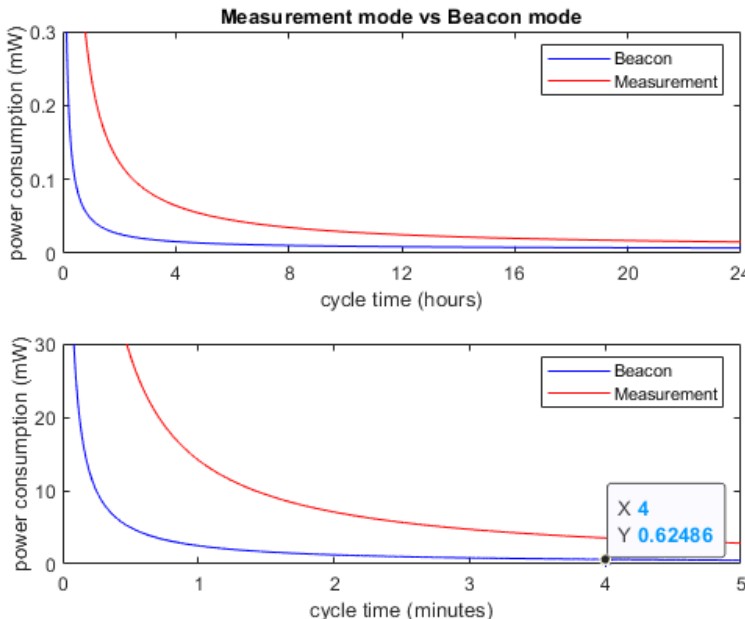

**Figure 12.** Sensor Node: average power consumption for the Measurement and the Beacon modes.

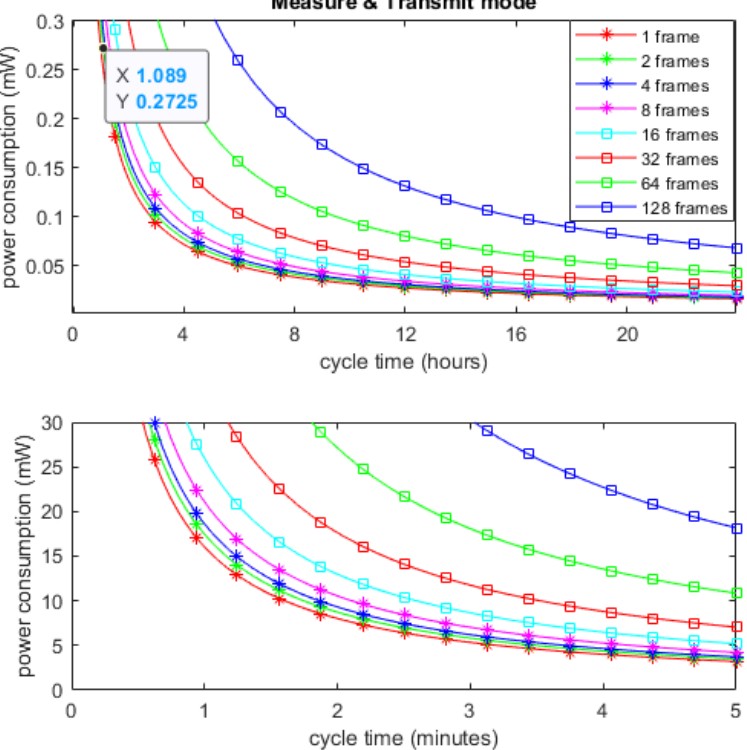

**Figure 13.** Sensor Node: average power consumption for the Measure & Transmit mode for different number of frames.

For the WATEREYE use case, if we consider that the Sensor Node remains in Measure & Transmit mode sending a single data frame on each tranmission, then, with a 20.88 Wh battery, by setting a cycle time of 1 h, a battery life of 8.7 years is estimated taking into

account the measured average power consumption of the sensor node for that cycle time as is highlighted in Figure 13. However, taking into account that batteries self-discharge over time, the real battery life-time will be shorter than the theoretical value. We have used the LSH14 lithium battery from SAFT which specifies a self discharge less than 3% after 1 year of storage at +20 °C.

*5.4. Anchor Measurements*

Table 4 shows the average power consumption and the time duration in the on-time and off-time for each type of mode.

**Table 4.** Anchor: average power consumptions and time durations for the different modes.

| Mode | $T_{ON}$ (ms) | $\overline{P_{ON}}$ (mW) | $\overline{P_{OFF}}$ (mW) |
|---|---|---|---|
| Beacon | 104 | 136.8 | 0.0054 |
| Ranging | 30 | 540 | 40.68 |

Figure 14 shows the average power consumption ($\overline{P}$) by the Anchor in the Beacon mode as a function of cycle time ($T_C$). As in the power profiles of the Sensor Node, the longer the cycle time, the lower the average power consumption. Therefore it can be seen that the curve tends to $\overline{P_{OFF}}$.

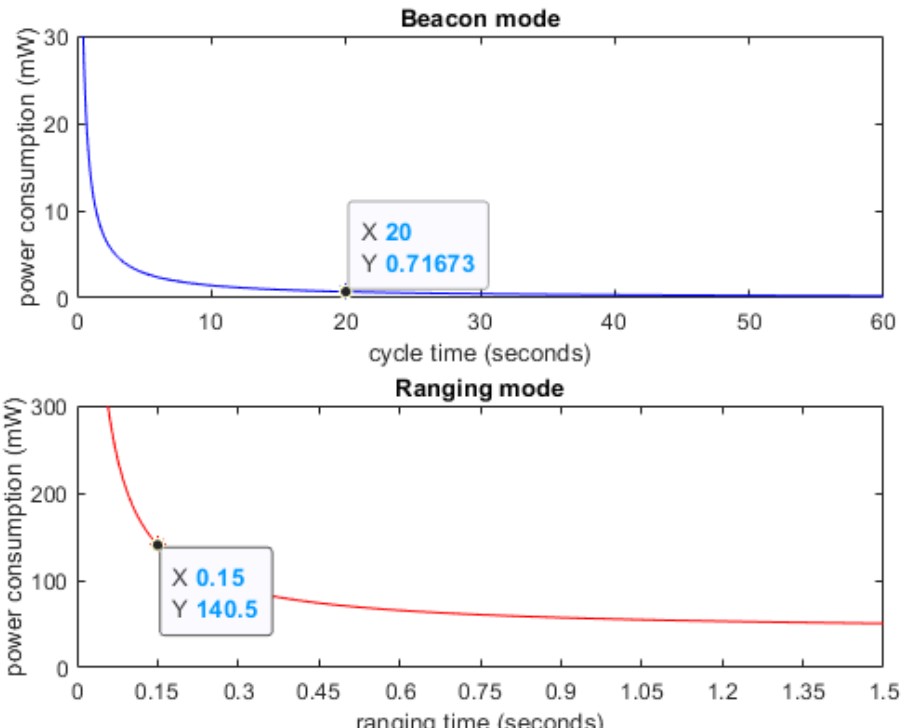

**Figure 14.** Anchor: average power consumption for the Beacon and the Ranging modes.

For the WATEREYE use case, if we consider that the Anchor will remain in Beacon mode during the whole battery life cycle, then, with a 20.88 Wh battery, by setting a cycle time of 20 s, a battery life of 3.3 years is foreseen.

Figure 14 shows the average power consumption ($\overline{P}$) by the Anchor in the Ranging mode as a function of ranging time ($T_R$). In this case, the longer the $T_R$, the lower the average power consumption. It can be seen that the curve tends to $\overline{P_{OFF}}$ as well.

For the WATEREYE use case, if we consider that the Anchor will remain in Ranging mode during the whole battery life cycle, then, with a 20.88 Wh battery, by setting a ranging rate of 150 ms, a battery life of 6.2 days is estimated. It can be seen that this is a high

consumption mode and therefore, it must only run when the Drone expressly requests ranging estimations during the flight.

Bearing in mind that corrosion is a slow process, the Drone will perform measurements once every several days. The UWB Anchors should be pre-configured to produce Beacon events with a small duty cycle (5–10 s) in those specific dates where the Drone is scheduled to fly. This way, the ranging requests from the Drone will be attended almost straight away. The rest of the days, the Anchors should increase their high duty cycle (1–6 h) for battery saving purposes.

## 6. Coverage and Packet Loss Measurements

The objective of the coverage and packet loss measurements is to analyze the range and reliability of the main communication link of the sensor nodes developed within the scope of the WATEREYE Project and its continuity in a given range by analyzing the loss of messages and gaps as a function of distance. For this purpose, the coverage in different environments is studied.

### 6.1. System Setup

A specific system architecture has been designed to perform the coverage measurements. The system consists of the following blocks: one Anchor, one Mobile Node and the WEC as can be seen in Figure 15. The behaviour of these components is not exactly the same as the expected one for the final system defined in Section 4.

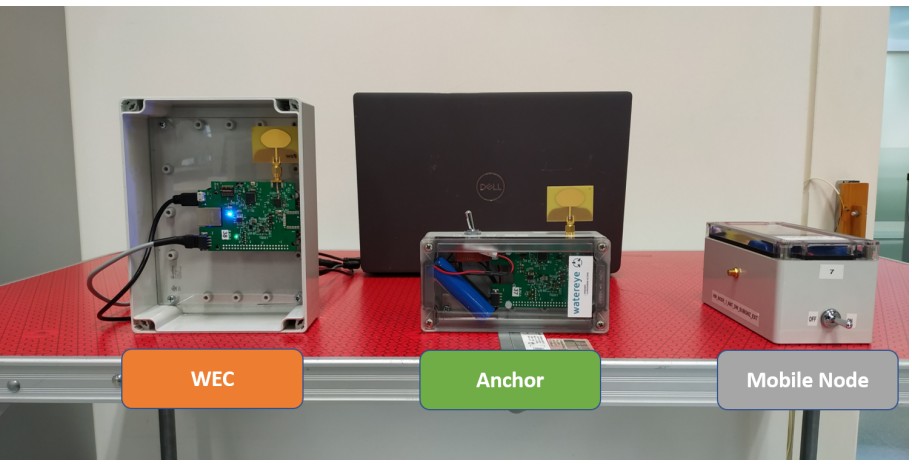

**Figure 15.** Implemented setup for the coverage measurements.

Once all the switches of the different blocks are ON, the system waits for the server developed by CEIT to start the operation. When this occurs, the WEC gives the order to the Mobile Node to start ranging with the Anchor. Once this starts, the WEC receives the information from the UWB ranging link and from the UWB communications link every 150 ms. It can be said that every 150 ms we obtain the status of the completed ranging (UWB—Channel ranging) and the status of the completed communication messages (UWB—Channel communications). On the other hand, the WEC stores the status data related to both UWB links for ranging and for communications with the aim of sending all the information to the Server through 3G in real time. In addition, the WEC checks the status of the Server every 60 s, and sends the information stored in its log to that server by 3G every 60 s.

### 6.2. Methodology

The objective of this section is to determine the coverage of the communications link and range link by analyzing the RSSI of the message exchange between Node/WEC and Node/Anchor. In addition, the range estimate made between Node and Anchor will be analyzed. For this purpose, it is necessary to validate that the communications link and the

ranging link can work correctly operating on different channels. To verify this, the system has been validated in different scenarios allowing us to analyze the system in key harsh environments.

The communication between the Node and the WEC is considered correct if a complete exchange of five messages is carried out, starting and ending at the Node. This exchange is performed every 150 ms. In the case of the Node and the Anchor, the communication is considered correct if a complete exchange of four messages is carried out. This exchange is initiated by the Node and terminated by the anchor. The exchange occurs every 150 ms as well.

### 6.3. Tests at Indoor Corridor: CEIT

Figure 16 shows the indoor environment located in a corridor of CEIT BRTA, which allows a maximum distance of 60 m for the tests.

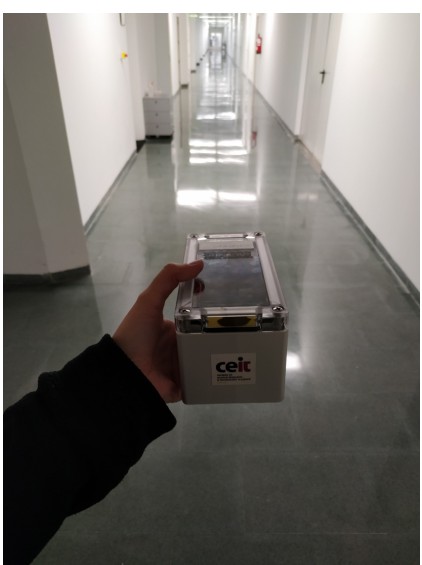

**Figure 16.** Test site for indoor coverage measurements at CEIT BRTA.

The anchor and the WEC are located on a fixed surface, in this case a table, while a person moves the sensor node. Each measurement starts at a distance of 1 m between the anchor and the WEC with respect to the sensor node, which is moved every 10 m, waiting one minute for each movement. A straight line route is made up to the maximum possible distance allowed by the different environments, remaining for 2 min at that maximum distance. Then, the mobile node returns in the same way in a straight line and waiting for the same 1 min in each position up to the starting position. The antennas of the equipment are kept facing each other and at the same height as far as possible.

Figure 17 shows the estimate of the distance between the anchor and the sensor node over time and its corresponding RSSI (see Figure 17b where the green curve represents the frame RSSI and the red curve, the RSSI of the first path). Both the frame RSSI and the RSSI of the first path are estimated by our system based on the raw information provided by the Decawave integrated chip. The first path RSSI must be always lower than the receive signal power in the entire frame. It can be seen that the distance increases as the sensor moves along the established path with respect to the anchor and decreases as the sensor node returns to the beginning of the path. The maximum distance of 60 m is reached.

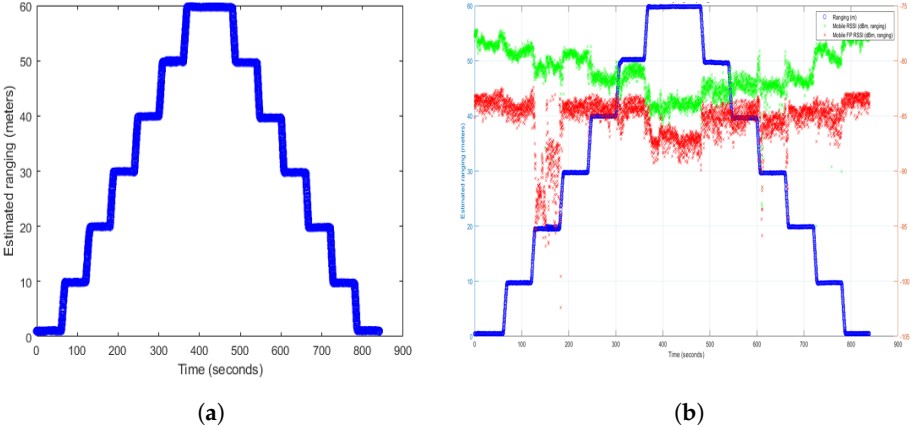

(**a**)             (**b**)

**Figure 17.** Estimated ranging and RSSI Test 1. (**a**) Estimated ranging of the UWB ranging link. (**b**) Estimated ranging of the UWB ranging link and RSSI of the UWB communications link.

Table 5 shows the results of the ranging and communication message exchange between the devices in an indoor corridor environment. It can be seen that the packet loss is low, so the reliability of both links is considered high.

**Table 5.** Results in test 1 of the amount of communications exchanged and ranging messages.

| Name | Maximum Range (m) | Duration (s) | Estimated Messages | Packet Loss (%) (UWB Communications Channel) | Packet Loss (%) (UWB Ranging Channel) |
|------|-------------------|--------------|--------------------|--------------------------------------------|--------------------------------------|
| Test 1 | 60 | 840.3 | 5602 | 1.4 | 0.7 |

*6.4. Tests at Indoor Harsh Environment: ELICAN Offshore Prototype*

The ELICAN prototype is a single offshore WT made of reinforced concrete as can be seen in Figure 18. This coverage test consists of deploying the system inside the ELICAN tower. This environment allows us a maximum distance of 60 m which is the height of the tower. During these tests, relevant data were collected for about 4 h to evaluate the communication link at different heights inside the ELICAN prototype.

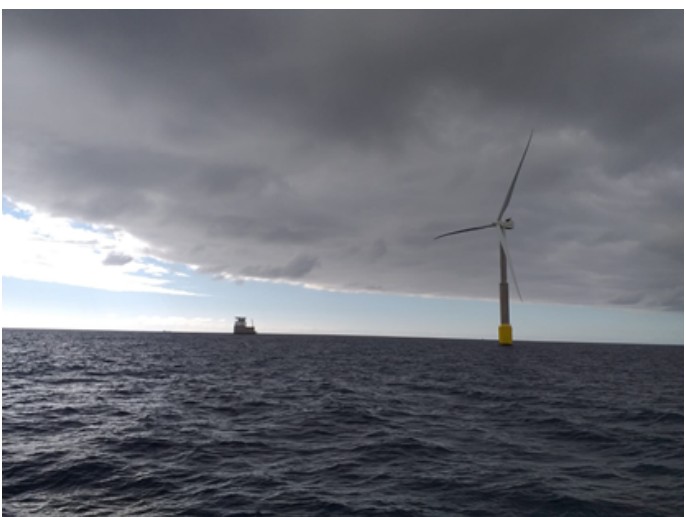

**Figure 18.** Photo of the ELICAN prototype.

Figure 19 shows how the equipment was placed inside the tower.

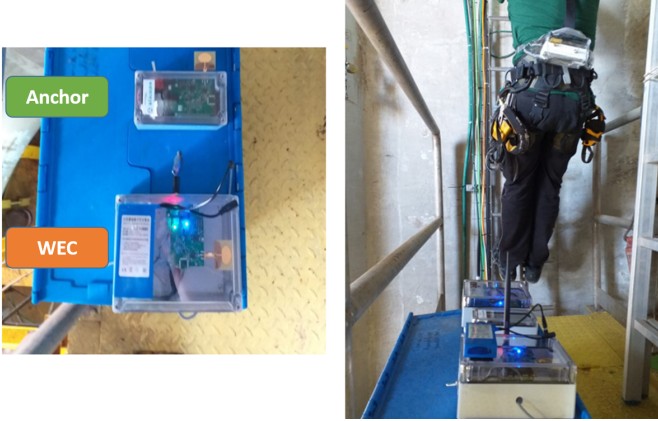

**Figure 19.** Communications system deployed inside the ELICAN prototype.

Figure 20 shows the layout of the ELICAN prototype and highlights the different positions where the worker waited the remote instructions from CEIT to continue with the test.

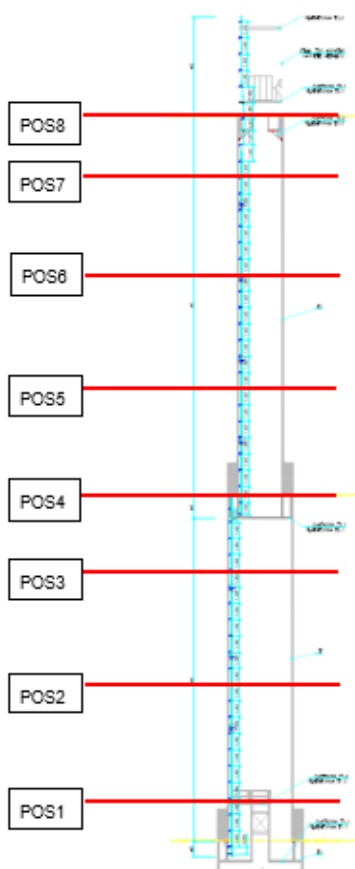

**Figure 20.** Communications system deployed inside the ELICAN prototype.

Table 6 shows the positions and their corresponding distances.

**Table 6.** Positions inside the ELICAN prototype.

| Position | Distance (m) |
| --- | --- |
| POS1 | +12 |
| POS2 | +21.8 |
| POS3 | +31.6 |
| POS4 | +36.92 |
| POS5 | +45.24 |
| POS6 | +55.04 |
| POS7 | +64.84 |
| POS8 | +69.92 |

The Anchor and the WEC were attached to the platform. The worker took the mobile node and climbed the tower with it from POS1 to POS8 (see Figure 21). Afterwards, he went down from POS8 to POS1. The worker remained in the same position for approximately 1 min. Actually, the timing in each position was remotely controlled by CEIT.

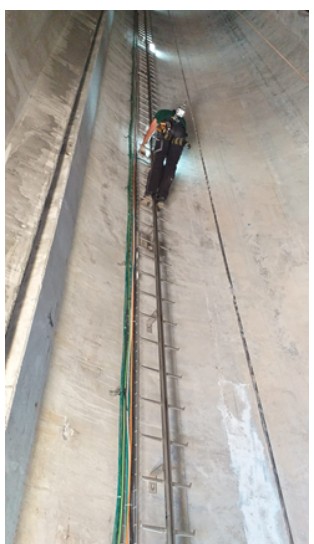

**Figure 21.** Mobile sensor node going up inside the ELICAN prototype.

Figure 22 shows the estimated range obtained when moving inside the tower. At the end of the test, it can be appreciated a slight void, only in the UWB ranging link, that may have been caused by the movement of the worker descending the ladder. In spite of that fact, it can be seen that it is a continuous movement and the system is able to follow the sensor node during the whole test. On top of that, as is shown in Figure 22b, if we focus on the RSSI results of the UWB communications link, although the frame RSSI and the RSSI of the first path are affected by the environment as expected, any gap of data can be seen except at the end of the experiment which was produced by the own worker.

Table 7 presents the results in terms of the packet loss for the two UWB channels used for the estimated ranging and for the communications, respectively. If we compare the channels, the configuration we used for the UWB communications channel achieved the best results, with a packet loss of 5.5% and a good performance in terms of continuity.

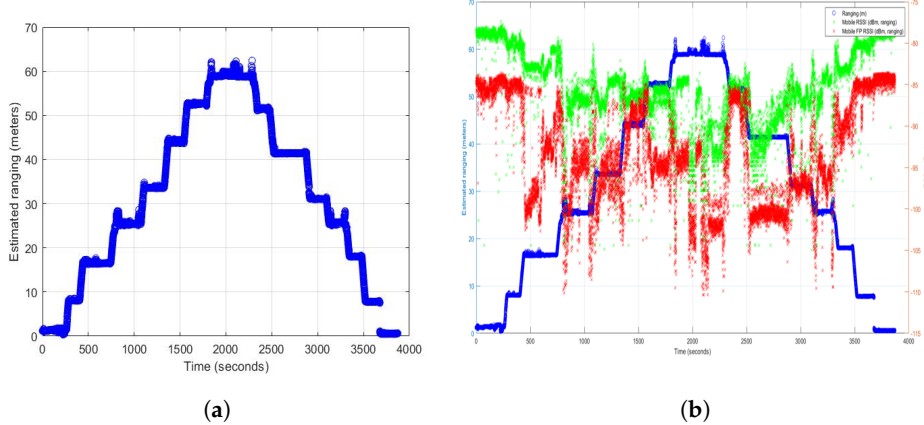

(a)    (b)

**Figure 22.** Estimated ranging and RSSI Test 2. (**a**) Estimated Ranging of the UWB ranging link. (**b**) Estimated ranging of the UWB ranging link and RSSI of the UWB communications link.

**Table 7.** Results of the coverage tests in the ELICAN prototype.

| Name | Maximum Range (m) | Duration (s) | Estimated Messages for Communications | Packet Loss (%) (UWB Communications Channel) | Estimated Messages for Ranging | Packet Loss (%) (UWB Ranging Channel) |
|---|---|---|---|---|---|---|
| Test 2 | 60 | 3598 | 17,300 | 5.5 | 17,462 | 9.5% |

### 6.5. Tests at Outdoor Harsh Environment: ArcelorMittal

The measurements in this test were carried out at the ArcelorMittal steel company located in Olaberría Guipuzkoa, specialized in the production of long steel products. It is considered a hostile environment because the environment can influence the operation and performance of the sensor system due to the metallic objects and electromagnetic fields. This site provided maximum distances of 100 m. Figure 23 shows the environment in which these measurements were made.

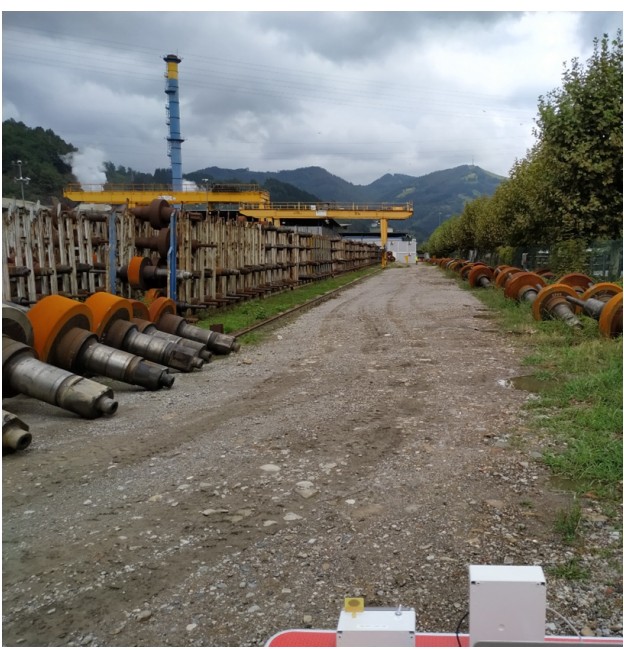

**Figure 23.** Test site for harsh environment coverage measurements at the ArcelorMittal steel company.

The procedure for these measurements is the same as the presented in the indoor corridor tests (Section 6.3). The devices were fixed on a table while the sensor node moved

in a straight line stopping every 10 m and waiting one minute at each position until it reached the maximum distance. The system waited two minutes at the maximum distance and started the same route back to the initial position.

Figure 24 shows the estimate of the distance between the anchor and the sensor node over time. It can be seen that the movement has been continuous and the maximum distance provided by this environment has been covered successfully.

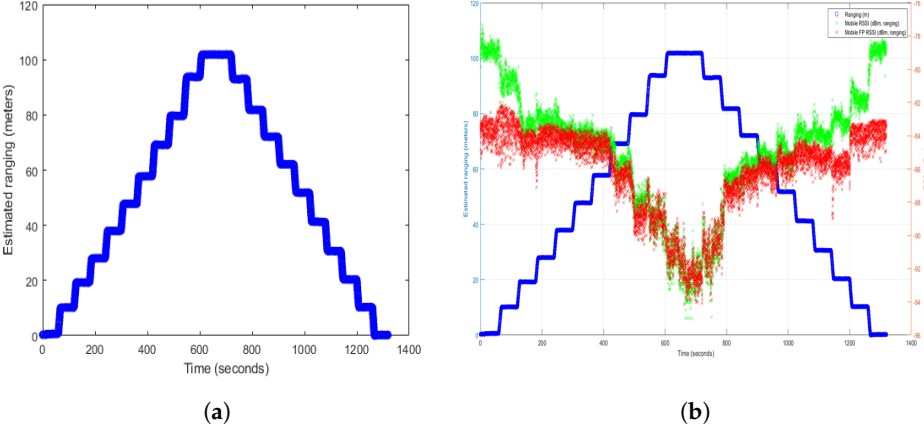

(a)                                                                 (b)

**Figure 24.** Estimated ranging and RSSI Test 3. (**a**) Estimated Ranging of the UWB ranging link. (**b**) Estimated ranging of the UWB ranging link and RSSI of the UWB communications link.

Table 8 shows the results of the ranging and communication messages in this outdoor harsh environment. A percentage of below 2% in terms of packet loss is achieved for the two UWB channels used for the estimated range and for communications.

**Table 8.** Results in test 3 of the amount of communications exchanged and ranging messages.

| Name | Maximum Range (m) | Duration (s) | Estimated Messages | Packet Loss (%) (UWB Communications Channel) | Packet Loss (%) (UWB Ranging Channel) |
|------|-------------------|--------------|--------------------|----------------------------------------------|----------------------------------------|
| Test 3 | 100 | 1319.2 | 8795 | 1.6% | 1.3% |

*6.6. Discussion of the Coverage and Packet Loss Results*

In the three tests carried out, it has been possible to verify that both the communications link and the ranging link have worked correctly, have not had interferences and have not stopped working. The packet loss in the corridor environment and the hostile environment of ArcelorMittal is below 2% for both links. This demonstrates the high reliability of the wireless solution. In the case of the offshore tower, the packet loss is higher (below 10%) due to the harshness of the environment (a real offshore wind turbine). However, that packet loss is fair enough to get the expected continuity for both the communications link and the ranging link. Additionally, the presented results show that the system is capable of following the different trajectories at low speed but complying with the timing requirement of 150 ms among the ranging estimations. In the case of the test inside the tower, it can be seen in Figure 22a that there is a slight void caused by the movement of the worker descending the ladder affecting the antenna orientation of the node carried by the worker. Furthermore, in Figure 22b both the frame RSSI and the RSSI of the first path are affected by the metallic objects inside the tower and the tower structure itself. In spite of that fact, the performance of the proposed wireless links are good based on the ranging estimations, the low packet loss and the continuity in the messages reception as is shown in Figure 22a,b.

## 7. Conclusions

To reach the ambitious renewable energy targets stated in [1], smart O&M strategies that lead to a levelized cost of energy reduction in OWF are required. Taking this into

account, this paper presents a new approach based on the design of a low-power wireless sensor network architecture for corrosion monitoring.

In the proposed architecture, low-power strategies have been implemented at both FW and HW levels in order to obtain autonomy for several years. On the one hand, the Sensor Nodes and the UWB Anchors employ sleep/wake-up mechanisms to minimize the average power consumption. The target autonomy is achieved thanks to the short on-times and the ultra-low current requirements during off-times. Furthermore, each wake-up event is synchronized in order to carry out scheduled, hence, collision-free, beacon, data and ranging communications. On the other hand, the presented HW design is power-efficient as is capable of enabling the components of the systems only when they are required for performing the tasks related to the corrosion measurements or communication events.

Different tests have been carried out to provide a deep power consumption analysis of the key components that compose the proposed architecture. Thus, the life-expectancy of the Sensor Nodes and the UWB Anchors has been estimated by varying the parameters of the sub-systems, such as the type of event, the duty cycle and the number of frames to be sent. Considering the WATEREYE use case, the presented power measurements show that the requirement for a battery autonomy of 3–5 years can be fulfilled by applying reasonable duty-cycles. Taking into account that corrosion is an inherently slow process, the Sensor Nodes can properly monitor the critical points of the structure by making measurements once every hour. In addition to that, the Nodes will hardly require to be synchronized/reconfigured in normal operation. Therefore, sending a beacon every 4 min, mainly as a keep-alive signal, is adequate. As far as the UWB Anchors, the duty-cycle will be adapted depending on the schedule of the Drone flights.

Furthermore, the presented coverage and packet loss measurements validate the proposed communication system, not only in a controlled scenario as an indoor corridor, but also in hostile environments as the surroundings of an steel company and specially, inside a real offshore wind turbine. The test results show that, at every analyzed scenario, the data and the ranging links can work together operating on different channels. More precisely, the analysis of the loss of messages and gaps as a function of distance shows that both links can work continuously in ranges up to 100 m. Therefore, considering the WATEREYE use case, the presented results proof that the requirement for covering an effective range of 80 m, and accordingly, the full extension of a wind turbine, can be fulfilled. In other words, once the whole WATEREYE system is deployed in a wind turbine, every Sensor Node will be within radio range of the WATEREYE Computer and every UWB Anchor will be within range of the Mobile Sensor Node, fulfilling the key requirement of the proposed double-star network.

**Author Contributions:** Conceptualization, A.C.; Formal analysis, A.C. and A.G.; Funding acquisition, A.C.; Investigation, A.C., A.G., J.C., P.B., A.d.S. and M.L.; Methodology, A.C., A.d.S., M.L. and A.G.; Project administration, A.C.; Software, P.B., A.G., J.C. and M.L.; Supervision, A.C. and P.B.; Validation, A.G., A.d.S., J.C. and M.L.; Writing—Original draft, A.G.; Writing—Review & editing, A.C. and P.B. All authors have read and agreed to the published version of the manuscript.

**Funding:** This work was supported by the WATEREYE project which has received funding from the European Union's Horizon 2020 research and innovation programme under grant agreement No. 851207.

**Acknowledgments:** This work has been possible thanks to the cooperation of CEIT with all the WATEREYE partners, especially in this case Delft Dynamics and PLOCAN.

**Conflicts of Interest:** The authors declare no conflict of interest.

## Abbreviations

The following abbreviations are used in this manuscript:

| | |
|---|---|
| ACK | Acknowledgement |
| ADC | Analogue-to-Digital Converter |
| BBB | BeagleBone Black |
| CoAP | Constrained Application Protocol |
| DDS | Drone Docking Station |
| DWRTT | Doble Way Round-Trip-Time |
| EU | European Union |
| FPGA | Field-Programmable Gate Array |
| GPIO | General Purpose Input-Output |
| HART | Highway Addressable Remote Transducer |
| HW | Hardware |
| IEEE | Institute of Electrical and Electronics Engineers |
| IC | Integrated Circuit |
| ID | Identification |
| I2C | Inter-Integrated Circuit |
| IP | Internet Protocol |
| ISA | Industry Standard Architecture |
| LCoE | Levelized Cost of Energy |
| LDO | Low-Dropout |
| O&M | Operations & Maintenance |
| OWF | Offshore Wind Farm |
| OWT | Offshore Wind Turbine |
| PLOCAN | Plataforma Oceanica de Canarias |
| PRU | Programmable Real-Time Unit |
| RTC | Real Time Clock |
| RTT | Round-Trip-Time |
| RUL | Remaining Useful Life |
| SHM | Structural Health Monitoring |
| SNMP | Simple Network Management Protocol |
| SPI | Serial Peripheral Interface |
| SRAM | Static Random Access Memory |
| ST | STMicroelectronics |
| SWD | Serial Wire Debug |
| TCP | Transmision Control Protocol |
| TI | Texas Instruments |
| ToF | Time-of-Flight |
| UART | Universal Asynchronous Receiver-Transmitter |
| USB | Universal Serial Bus |
| UWB | Ultra Wide-Band |
| WEC | WATEREYE Computer |
| WSN | Wireless Sensor Network |
| WT | Wind Turbine |

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
