# Peer review of "Reliable and Low-Power Communications System Based on IR-UWB for Offshore Wind Turbines"

_electronics, doi:10.3390/electronics11040570_

Round 1

Reviewer 1 Report

A few minor revisions:

  1. Page 1, line 4, change "composed by" to "composed of"
  2. Page 2, lines 51,52, use the greek symbol "mu" instead of 'u"
  3. Page 9, line 314, change "an" to "a"
  4. Figure 8, increase the font size
  5. Page 12, after Equ (1), Correct the spelling of "Double"
  6. Page 15, line 427, change "influenciated" to "influenced"
  7.  

Reviewer 2 Report

In this paper authors present the end-to-end design and development of a system based on embedded systems and open low power communication technologies able to continuously monitor offshore wind turbines in order to maintain it against corrosion.

The work is certainly interesting, relevant to the specific journal and of significant added value as the result of a EU project.

On the positive aspect it is very nice to see a complete system/prototype tested in real environment. Additionally, the paper presents all the critical aspect required of such a system.

The overall system architecture is convincing although not especially innovative, but probably the feasibility and performance is the most important aspect.

Also, the hardware designs presented and the algorithms in relation to energy conservation and pre-processing is also accounted in the positive points of the paper.

Finally, the use of English is adequate although a care full review from an native English speaker would be useful to improve the wording in some cases.

However, there are some aspects that raise concern and should be adequately addressed or elaborated as follows…

Although low power consumption is in general something positive and desirable it is not clear why this is important in the case of a huge wind turbine which obviously can provide the required energy for many years without a problem. Or in such deployments it is relatively easy to use low cost energy harvesting components such as small solar panel with batteries to power such low power embedded system as the ones described. So the significance of low power in this use case should be better argued.

Related work is limited as to the sensoring part, data processing part and some very well known SotA communication technologies are not included such as LoraWAN, 4G/5G setups etc.

Elaboration is required in relation to “where the marine air conditions and the action of tides and waves must be taken into account. These conditions affect the operation of the wireless links and new strategies must be proposed to improve their performance” indication which is not very clear.

Authors should elaborate and analyze on specific modalities that are monitored to measure corrosion and the levels considered.

Please provide more information on the IEEE 802.15.4 wireless interfaces used. The indicated throughputs of 6800kbps seem quite excessive for the specific protocol. Information on how is this measured is very important here?

Concerning sensor node board and UWB Anchor board.: Why do you use a custom-made board and not an already existing one? How does this compare to COTS existing solutions with the same components? The IEEE 802.1.5.4 wireless interface design and performance must be elaborated. Please provide information on which interface do you use, which network stack etc.

“Doble Way Round-Trip-Time” should be “Double Way Round-Trip-Time”.

Figure 9 should be accompanied by a real picture of the real deployment.

How is the “average power consumption for each type of event” calculated should also be elaborated since it comprises from segments of different duration and power consumptions? Please elaborate.

Figure 11a should indicate the exact measurements and the units of them.

Please analyze how 3.8 years and 8.7 years are estimated in page 16.

Considering Figure 17b and the indication of RSSI of the first path. Please elaborate on what you mean and how you distinguish the first from the rest of the paths's RSSI.

Considering the overall performance evaluation 1) It would be very helpful and convincing as to the efficiency of the deployment, if some measurements were made of a period of several days or e.g. 1-2. weeks so that the specific measurements presented here as well as the power consumption measurements and the expected lifetime estimations could be verified and 2) measurements as to the communication performance e.g. maximum throughput, max delay etc. in such very interesting harsh environments would be extremely useful.

Finally, probably the most important aspect that the reviewer thinks it is strange and doesn’t agree with respect to the initial specifications of the system are the wireless communication technologies selected. Technologies indicated such as IEEE 802.15.4, IR-UWB, ZigBee etc. are somehow outdated and don’t seem adequate for such harsh environments. Currently many other solutions that at first glance would seem a better fit exist such as LoraWAN, Bluetooth, Bluetooth Low Power and even WiFi. Bluetooth and BLE in many paper is used as an alternative to IEEE802.15.4/ZigBee with better low power feature and analogous coverage, limited, performance. WiFi modure exist nowadays for ultra low power embedded systems such as raspberries, Arduinos and ARM based platforms featuring very low power consumption performance and much better throughput and delay characteristics. Finally, since the spec of the application require low traffic, extended coverage and low power requirements LoraWAN seems ideal for this system. And as already indicated ultra-low power doesn’t seem critically important when residing in wind turbine installations. Authors are required to accurately argue on the wireless technology selection as opposed to existing solutions.

Summarizing the work presented is very interesting and a lot of effort seems to have been devoted to implement and test the end system. And this has significant merit. But there are several points that raise concerns tha authors must argue, elaborate, extend or correct for this work the be accepted for publication.

Reviewer 3 Report

This paper is well written and shows the usage of a communications system tested for a specific application. Totally it is acceptable but the following items should be addressed in the revised version of the paper or to be answered.

1. In the title of the paper, we can see the “Reliable.” How much reliability is it? Is it measured by outage probability, packet loss probability, or other metrics? It should be carefully described.

2. In lines 243, 244, “Get a throughput of 150ms mainly for the ranging estimations to provide the capability of following a trajectory.” Should be revised. Throughput is a well-known metric in communications in (bit/s) or (bit/s/Hz).

3. In lines 226-246, you want to show the effectiveness of the IR-UWB but for some other methods and technologies such as CDMA, WCDMA, OFDM, … maybe we have better performance. I think this question “Why IR-UWB is selected?” should be carefully answered. In your answer, other systems should be mentioned and your comparisons should be summarized in a table.

4. The lack of a table to show the parameters, descriptions, and values is obvious. For example, the receiver threshold, noise power or noise power spectral density, center frequencies, bandwidths, …

5. You have large bandwidth but I think the data rates required for this application are not high. It must be detailed to clearly show the bit rate requirements.

6. In addition to the consumed power and coverage metric, the well-known metrics to evaluate the reliability and performance should be used. I suggest the authors use packet loss or BER, or SER instead of the received messages in Tables 6, 7.

7. Some graphs to compare the simulation results and measurements are needed to show the validity of your work.

8. I suggest the authors study some related papers and the results and comparisons in the Introduction and related work parts.

Reviewer 4 Report

Fig. 12 (b) is shown with a data point, why? The legend is also missing here. Also, show (a) and (b) in this figure. 

Fig 13 (a) should be re-drawn because the legends are covering the results. Try to fix the legends like a two-column instead of 1 column. 

Sec. 6.6 should be extended. It is inconclusive discussion. 

Also, add a parametric table covering all the parameters utilized in this work. with specific details.   

In Figure 10, the desktop app is very blurred and is not explained. 

Fig. 8 and 11 should have neutral colors. 

Fig. 22 (b) has not been explained. The image is also not clean. it should be well placed with a justified explanation

Round 2

Reviewer 2 Report

The authors responded to the reviewer’s concerns satisfactorily in most cases and the respective clarifications and elaborations are welcome.

Several concerns are not fully covered but it is recognized that they are initiated by original specifications taken in the context of the project (rightfully or not). In the project overall as indicated in the initial review significant and substantial effort and work has been undoubtedly devoted. So, the work presented overall can be of interest or a wide range of audience, so this balances out some concerns that remain from the reviewer side.

No more comments.

Reviewer 3 Report

The revised version of the paper satisfies the reviewer because it covers the requirements that I mentioned in the previous review.

Reviewer 4 Report

Comments have been addressed.